# Recruiting transgender men in the Southeastern United States for genital microbiome research: Lessons learned

**Olivia T. Van Gerwen**[1]*, **Z. Alex Sherman**[1], **Emma Sophia Kay**[2], **Jay Wall**[3], **Joy Lewis**[1], **Isaac Eastlund**[1], **Keonte J. Graves**[1], **Saralyn Richter**[1], **Angela Pontius**[1], **Kristal J. Aaron**[1], **Krishmita Siwakoti**[4], **Ben Rogers**[5], **Evelyn Toh**[6], **Jacob H. Elnaggar**[7], **Christopher M. Taylor**[7], **Nicholas J. Van Wagoner**[1], **Christina A. Muzny**[1]

1 Division of Infectious Diseases, University of Alabama at Birmingham, Birmingham, Alabama, United States of America, 2 School of Nursing, University of Alabama at Birmingham, Birmingham, Alabama, United States of America, 3 Magic City Wellness Center, Birmingham, Alabama, United States of America, 4 Division of Endocrinology, University of Alabama at Birmingham, Birmingham, Alabama, United States of America, 5 University of Alabama at Birmingham Heersink School of Medicine, Birmingham, Alabama, United States of America, 6 Department of Microbiology and Immunology, Indiana University School of Medicine, Indianapolis, Indiana, United States of America, 7 Department of Microbiology, Immunology, and Parasitology, Louisiana Health Sciences Center New Orleans, New Orleans, Louisiana, United States of America

* oliviavangerwen@uabmc.edu

**Data Availability Statement:** We have attached a spreadsheet in this submission containing the raw, deidentified data.

## Abstract

### Background

Transgender men (TGM) are underrepresented in genital microbiome research. Our prospective study in Birmingham, AL investigated genital microbiota changes over time in TGM initiating testosterone, including the development of incident bacterial vaginosis (iBV). Here, we present lessons learned from recruitment challenges encountered during the conduct of this study.

### Methods

Inclusion criteria were assigned female sex at birth, TGM or non-binary identity, age ≥18 years, interested in injectable testosterone but willing to wait 7 days after enrollment before starting, and engaged with a testosterone-prescribing provider. Exclusion criteria were recent antibiotic use, HIV/STI infection, current vaginal infection, pregnancy, or past 6 months testosterone use. Recruitment initiatives included community advertisements via flyers, social media posts, and referrals from local gender health clinics.

### Results

Between February 2022 and October 2023, 61 individuals contacted the study, 17 (27.9%) completed an in-person screening visit, and 10 (58.8%) of those screened were enrolled. The primary reasons for individuals failing study screening were having limited access to testosterone-prescribing providers, already being on testosterone, being unwilling to wait 7

**Funding:** OTVG has received funding to her institution for research from Gilead Sciences, Inc., Moderna Pharmaceuticals, Visby, and Abbott Molecular and has served on a scientific advisory board for Scynexis, for which she received honoraria. CAM reports receiving grants to her institution from NIAID, Lupin, Abbott Molecular, Visby, and Gilead Sciences, Inc. She also reports honorarium and/or consulting fees from Scynexis, Cepheid, BioNTech, Visby Medical, Elsevier, UpToDate, Abbott Molecular, and Roche. All other authors have no disclosures. This study is funded by the National Institute of Allergy and Infectious Diseases (R21AI167754-01 granted to CAM, NVW, and CMT). OTVG is currently funded by a 2022-2024 Frommeyer Career Development Award from the UAB Department of Medicine.

**Competing interests:** The authors have declared that no competing interests exist.

days to initiate testosterone therapy, or desiring the use of topical testosterone. Engagement of non-White TGM was also minimal.

## Conclusion

Despite robust study inquiry by TGM, screening and enrollment challenges were faced including engagement by TGM not yet in care and specific study eligibility criteria. Excitement among TGM for research representation should be leveraged in future work by engaging transgender community stakeholders at the inception of study development, particularly regarding feasibility of study inclusion and exclusion criteria, as well as recruitment of TGM of color. These results also highlight the need for more clinical resources for prescribing gender-affirming hormone therapy, especially in the Southeastern US.

## Introduction

Transgender men (TGM) taking testosterone for gender-affirming hormone therapy (GAHT) who have not had bottom surgery can experience negative sexual health outcomes due to the impact of testosterone on the vaginal (henceforth referred to as genital) microbiota [1]. Specifically, testosterone can thin the genital epithelium and lead to atrophic vaginitis [2, 3]. One cross-sectional study of 28 TGM on testosterone for at least one year found that testosterone impacts the composition of the genital microbiota, depleting protective *Lactobacillus* species and supporting the overgrowth of diverse bacterial communities. These events can lead to genital dysbiosis and the development of incident bacterial vaginosis (iBV) [3]. This is consequential as iBV not only yields bothersome symptoms such as discharge, odor, and irritation [4], but is also a risk factor for the acquisition of HIV and other sexually transmitted infections (STIs) [5].

Despite a disproportionate focus on HIV/STI outcomes among transgender women due to high prevalence of these infections in this population, TGM, particularly those who have sex with men, are at increased risk of acquiring HIV and STIs compared to the general population [6, 7]. In fact, data from the 2009–2014 Medical Monitoring Project (MMP) estimate that one out of ten transgender HIV-positive patients were TGM, with 15.4% of all newly diagnosed HIV cases among transgender persons being TGM [8]. Reasons for this disparity are myriad, but include diverse sexual behaviors, systemic misogyny and transphobia, and poor engagement in care [9]. TGM living with HIV also have worse outcomes than other HIV-positive populations; in the MMP, 47% of TGM living with HIV were living in poverty, 23% had depression, and only 60% were able to sustain viral suppression [8]. As such, the implications of impacts of testosterone on the vaginal microbiota of this population have the potential to exacerbate the already poor sexual health outcomes this population faces.

Our team conducted a prospective, observational study with intensive genital specimen collection pre- and post-testosterone initiation to better understand the impact of testosterone on the genital microbiota among TGM over time, including the development of iBV. The protocol for this study, including extensive details of the methods and study rationale, was previously published [10]. TGM are significantly underrepresented in vaginitis and genital microbiome research, highlighting the need to conduct this type of research [11]. The reasons for this underrepresentation are not completely understood, but likely similar to those driving sexual health disparities in this population, including institutional mistrust, limited interest in

sexual health outcomes from researchers, and systemic misogyny and transphobia [12, 13]. Throughout this study, we closely monitored recruitment data to better understand how to engage this priority population in research studies. Here, we aim to present our team's recruitment strategies for this study, the socio-demographics of screened and enrolled participants, and describe the lessons learned from our recruitment challenges to help inform the design of future studies engaging underserved populations such as TGM in genital microbiome research.

## Methods

From the inception of this study, TGM in the Birmingham, AL community, as well as local providers and organizations serving TGM, were integral to the development of recruitment strategies utilized. From the community perspective, author JW provided insight as a TGM and community advocate and author ESK advised as a former community researcher embedded within a local LGBTQ+ health service organization, the Magic City Wellness Center (MCWC). Additionally, authors OTVG, NVW, and KS (physicians providing gender-affirming hormone therapy to TGM at the University of Alabama at Birmingham [UAB]) as well as Dr. Scott Weisberg (a physician at the MCWC) also advised on potential recruitment strategies from their perspectives as local trans care providers. These providers and establishments are all well-respected and trusted by the local transgender community. Additionally, all staff at these clinical locations have undergone cultural competency trainings, which have yielded welcoming and affirming care environments for patients. All members of the study team at the UAB Sexual Health Research Clinic (SHRC), where this study took place, also participated in the UAB Safe Zone training. This interactive, facilitated training course was tailored to enhance awareness of LGBTQ+ language, vocabulary, concepts and cultural sensitivity to promote a culture of safety and quality for providers and patients. The two primary methods of participant recruitment were 1) referrals from testosterone-prescribing providers at local gender health clinics and 2) a study flyer which was canvased in establishments frequented by the local TGM community (e.g., clinic websites, bars, breweries, restaurants, coffee shops, parks, cannabidiol dispensaries, venues) and on social media (e.g., Facebook, Instagram) [10].

In Birmingham AL, there are three major healthcare establishments that prescribe testosterone to TGM: the MCWC, the University of Alabama at Birmingham (UAB) Gender Health Clinic (GHC), and the UAB Student Health Services Sexual Health Clinic (UAB SHC). Testosterone-prescribing providers as well as mental health providers at these clinics were educated on the eligibility criteria for this study and invited to refer potential participants to complete an in-person screening visit at the UAB Sexual Health Research Clinic (SHRC). Providers could either offer potential participants a study flyer or have the participant call the UAB SHRC while in the room with the potential participant to expedite a study screening visit. Importantly, providers emphasized that if they enrolled, participants had to be willing to 1) wait 7 days before starting testosterone (to collect pre-testosterone genital specimens to establish a baseline for the genital microbiome prior to starting T) and 2) start on an intramuscular (IM) or subcutaneous (SQ) formulation of testosterone. While providers shared this information with patients, it is important to note that they never did so in a coercive manner and, regardless of the patient's interest in research participation, provided medical care in keeping with current standards of care. Despite the multiple preparations and dosing strategies available for testosterone for GAHT, this study only included participants who were willing to start on testosterone cypionate 50mg IM or SQ weekly to avoid pharmacokinetic differences between the different routes of testosterone which may impact pre-and post-testosterone genital microbiome study results [1, 14].

Based on the number of local TGM in care in Birmingham at the time the study proposal was written, we determined that we needed to enroll 48 TGM with normal baseline vaginal microbiota. Normal vaginal microbiota was defined as having no Amsel criteria [15] and a Nugent score 0–3 [16] (Table 1). Given that we expected that ~10% (n = 4) would have an STI at baseline [11] and an additional 10% (n = 4) would become lost to follow-up [17], we were left with 40 TGM to be included in the final analysis. In a prior study of cisgender women, ~55% of participants had normal vaginal microbiota at baseline [18], thus we proposed to screen an estimated 86 TGM to enroll 48 with normal vaginal microbiota.

Once participants learned about the study and called the UAB SHRC, they were phone screened to assess screening eligibility criteria (Table 1). While performing phone screening, SHRC staff collected details regarding the number of participants who qualified for the study and made an in-person screening appointment, the number who declined to participate, and the number who were ineligible for an in-person screening visit. SHRC staff also collected information about how participants learned about the study and if they did not schedule an in-person screening visit, the primary reason why they did not. Staff recorded these details in a de-identified Microsoft Excel spreadsheet. Descriptive statistics were subsequently performed on these recruitment data using Microsoft Excel.

Participants presenting for an in-person screening appointment that met study enrollment criteria were invited to enroll in the study [10]. If participants chose to enroll, they were educated that the study would last for 97 days (7 days pre-testosterone initiation and 90 days post-testosterone initiation) and that they would self-collect 3 daily genital swabs, which were used for vaginal Gram stain for Nugent scoring, 16S rRNA gene sequencing, and shotgun

**Table 1. Study eligibility criteria.**

| Screening Eligibility Criteria | |
|---|---|
| *Inclusion Criteria* | *Exclusion Criteria* |
| ≥18 years of age | Oral or intra-genital antibiotic use in the preceding 14 days |
| Assigned female sex at birth | Self-report of HIV infection |
| Presence of natal genitalia (i.e., vagina) | Pregnancy |
| Self-identify as transmasculine, non-binary, or genderqueer | Use of testosterone in the past 6 months |
| Desire to initiation intramuscular testosterone, but willing to wait 7 days after enrollment before starting | |
| **Enrollment Eligibility Criteria** | |
| *Inclusion Criteria* | *Exclusion Criteria* |
| No Amsel criteria* | Trichomoniasis on genital wet mount |
| Nugent score** of 0–3 with no *Gardnerella vaginalis* morphotypes | Symptomatic genital yeast infection |
| | Positive chlamydia, gonorrhea, or trichomoniasis NAAT at baseline |
| | Positive HIV test |

*Amsel criteria [15]: Clinical diagnosis of BV by Amsel criteria requires at least three of the following four criteria: (1) Homogeneous, thin white/gray discharge coating the walls of the vaginal, (2) ≥20% clue cells (e.g., vaginal epithelial cells studded with adherent bacteria) per high power field (hpf) on microscopic examination of vaginal fluid, (3) pH of vaginal fluid >4.5, (4) a fishy odor of vaginal discharge after addition of 10% KOH (i.e., the whiff test)

** Nugent score [16]: Microbiologic diagnosis of BV diagnostic using a vaginal Gram stain to determine the amount of lactobacilli (i.e., long gram-positive rods), small gram-negative and gram-variable rods (i.e., *G. vaginalis* or *Bacteroides* species), and curved gram-negative rods (i.e., *Mobiluncus* species) characteristic of BV. A Nugent score of 0–3 is consistent with a *Lactobacillus*-predominant vaginal microbiota, 4–6 with intermediate microbiota, and 7–10 with BV.

metagenomic sequencing [10]. In addition, participants were asked to complete a one-page daily diary collecting sexual behavior, sexual partner characteristics, and genital symptom data while in the study. Data from these daily diaries were essential to include in this microbiome research study as events such as sex, douching, and menses can have a major impact on the composition of the genital microbiome over time [17, 19, 20]. Enrolled participants subsequently returned all study materials to the UAB SHRC every 1–2 weeks, at which time they received the next week's supplies and a monetary incentive. Participants received $30 cash for screening, $25 for enrollment, and $30 per week for 13 weeks at sample drop off ($475 total for completing the entire study).

This study was approved by the University of Alabama at Birmingham Institutional Review Board (IRB), protocol # IRB-300008073. All participants who presented to SHRC for an in-person screening provided written informed consent with a member of the study team for the screening visit and if they were eligible for enrollment, they provided additional written informed consent for enrollment in the study. We did not include minors in this study.

## Results

Recruitment for this study occurred from February 1, 2022 to October 5, 2023, during which time 61 individuals inquired about participation. Enrollment stopped when the two-year grant funding period for this study ended. Details regarding how and where these potential participants learned about the study are shown in Fig 1. Of these 61 individuals, 16 (26.2%)

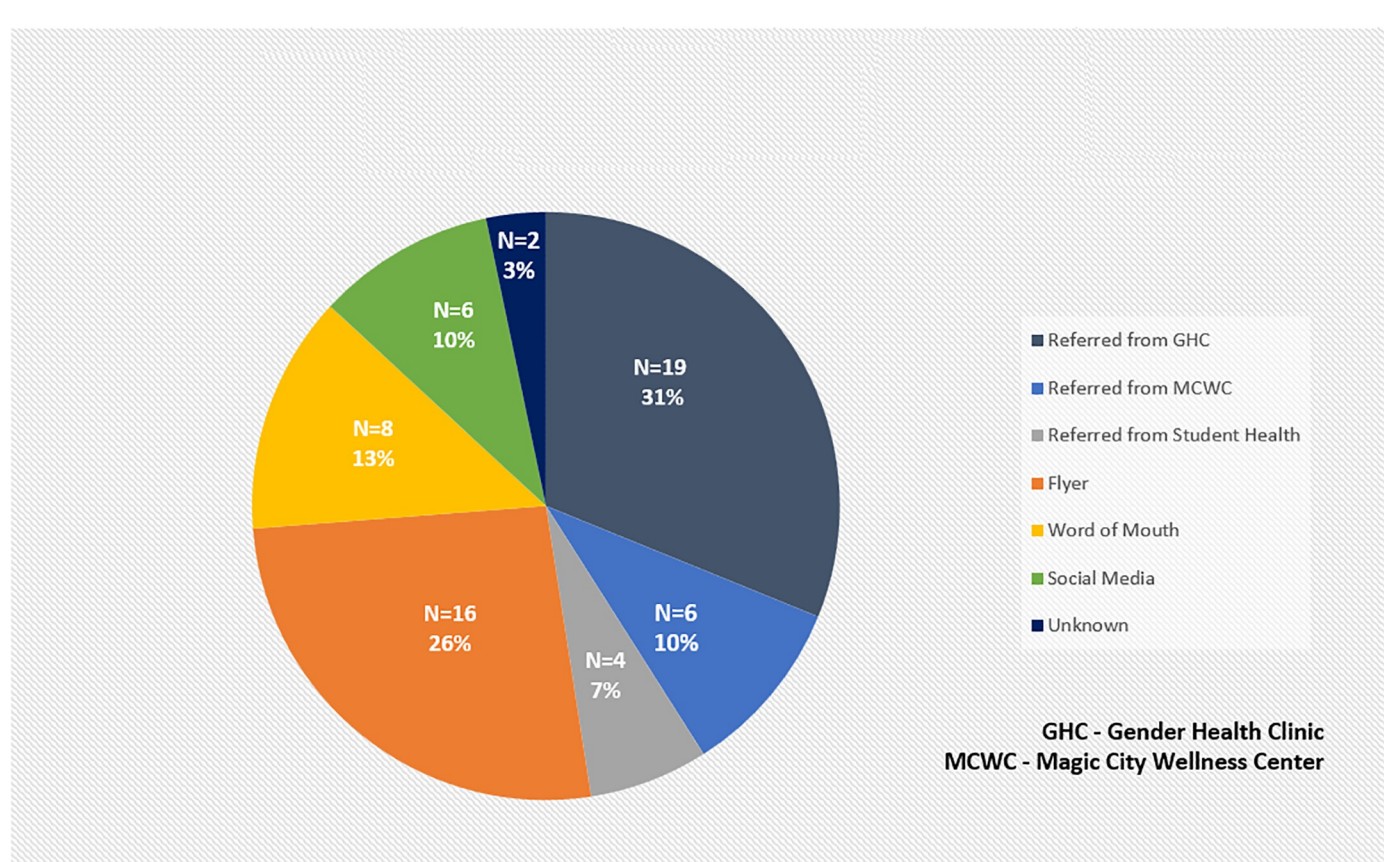

**Fig 1. Referral sources of potential participants who contacted the UAB Sexual Health Research Clinic about the study (n = 61).**

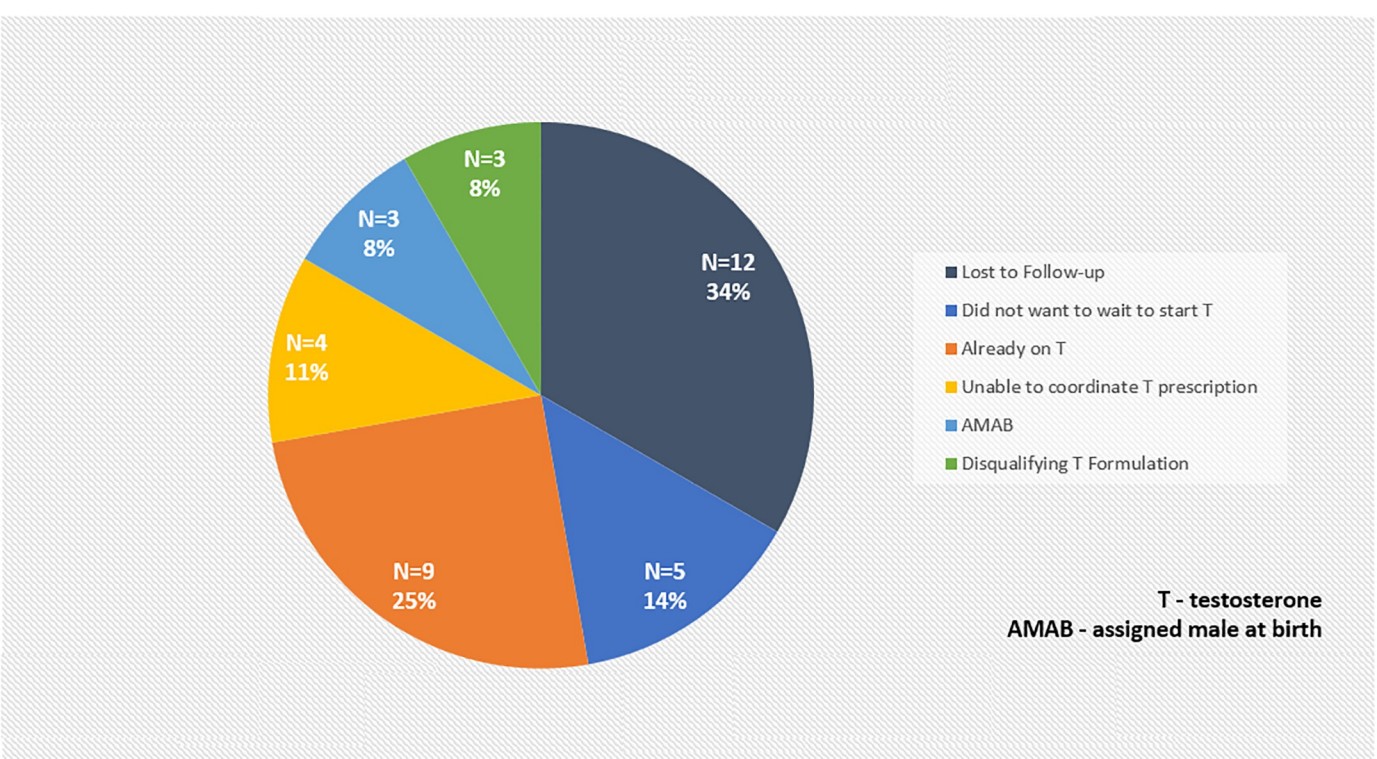

**Fig 2. Reasons that potential participants failed to complete an in-person screening (n = 44).** * Other reasons included the following: living too far away from the clinic (2; 4.5%), contacting the clinic after enrollment had closed (2; 4.5%), not interested in participatint (2; 4.5%), a medical condition precluding testosterone GAHT (1; 2.3%), and concerns about dysphoria related to vaginal swab.

responded to study flyers, 6 (9.8%) to social media postings, 8 (13.1%) due to word-of-mouth, and 19 (31.1%), 4 (6.6%), and 6 (9.8%) were referred from the UAB GHC, UAB SHC, and MCWC clinics, respectively. Two individuals (3.3%) did not share how they heard about the study. Seventeen of these 61 individuals (27.9%) qualified for and completed an in-person screening, with one of them rescreening after initially screen failing. Common reasons for the other individuals (n = 44) to inquire about the study but not complete a screening visit are shown in Fig 2. These included being lost to follow up (12; 27.3%), already being on testosterone for GAHT (9; 17.8%), not willing to wait 7 days to start testosterone (5; 11.4%), difficulty coordinating their testosterone prescription with a local health care provider (4; 9.1%), desiring or being prescribed a non-IM/SQ testosterone formulation (e.g., gel) (3; 6.8%), being assigned male sex at birth (3; 6.7%), being uninterested in participating in a research study (2; 4.5%), living too far away from the clinic (2; 4.5%), calling after the enrollment window had already closed as they had seen flyers that were still out in the community (2; 4.5%), concerns about dysphoria related to vaginal swab self-collection over a 97 day period of time (1; 2.3%), or a medical condition precluding testosterone GAHT initiation (1; 2.3%). When collecting data regarding the reasons for exclusion, only one option was documented for each individual.

The socio-demographic characteristics of the 17 individuals who qualified for and completed an in-person screening are shown in Table 2. Sixteen (9.41%) of those screened were white, non-Hispanic and one (5.9%) was white, Hispanic. Self-reported gender identity varied, with 11 (64.7%) individuals identifying as transgender men, 3 (17.6%) as non-binary, and 3 (17.6%) as genderqueer. Of the 17 screened, 10 (58.8%) individuals were enrolled and 7

**Table 2. Characteristics of screened and enrolled participants.**

| | N (%) or mean ± standard deviation | |
| --- | --- | --- |
| | Total screened in-person (n = 17) | Total enrolled (n = 10) |
| Age | 23±3 | 22±2 |
| Race/Ethnicity | | |
| White/non-Hispanic | 16 (94.1%) | 10 (100%) |
| White/Hispanic | 1 (5.9%) | 0 (0%) |
| Gender Identity | | |
| Transgender man | 11 (64.7%) | 8 (80.0%) |
| Non-binary | 3 (17.6%) | 1 (10.0%) |
| Genderqueer | 3 (17.6%) | 1 (10.0%) |

(41.2%) individuals were excluded either due to ≥1 Amsel criteria and/or a Nugent score >3 at screening.

## Discussion

To our knowledge, this is the first prospective study to enroll TGM in the Southeastern US to investigate the genital microbiota pre- and post-testosterone initiation, including the development of iBV. A primary strength of this study is the lessons learned from our recruitment challenges, which have allowed our team to identify areas of opportunity in improving TGM recruitment and engagement in future research studies. When planning for this study, we anticipated some degree of recruitment challenges given the systemic, community, interpersonal, and personal barriers faced by this population regarding research engagement and otherwise [12, 21]. These barriers as well as other recruitment challenges, summarized in Table 3, led us to screen and enroll fewer individuals than we aimed to do. In Table 3 and throughout this discussion, we present potential solutions to these challenges that can be leveraged in future studies by investigators doing similar work.

### Limited engagement with TGM not in care

While the response to study flyers posted around the community and on social media was robust, the number of individuals who desired or qualified for in-person screening were

**Table 3. Challenges and potential solutions for recruiting transgender men for sexual health research.**

| Challenge | Potential solutions |
| --- | --- |
| Limited engagement with TGM not in care | • Robust engagement in study design and implementation by TGM, especially those of color, rooted in the community<br>• Hiring and paying key popular opinion leaders to help inform study design and implementation<br>• Leveraging social media and geosocial networking applications (e.g., dating apps) for robust study advertising<br>• Including TGM in a variety of community-engaged research agendas to fulfill some of their unmet healthcare needs |
| Variations in available GAHT routes and doses | • Consideration for inclusion of all types of GAHT in future studies including TGD people |
| Dysphoria associated with genital specimen collection | • Flexibility in sample collection method and frequency<br>• Formative qualitative work to better characterize acceptability and feasibility of genital specimen self-collection for vaginal health evaluation |
| Inflexible study design | • More flexibility in eligibility criteria in future studies |

limited. One major reason for not qualifying was that many potential participants were already on testosterone for GAHT. At the time of this writing, approximately 400 TGM are in care at the three gender health clinics in our area and many of them are already on testosterone therapy. Future studies investigating the genital microbiota around the time of testosterone initiation will need to identify unique ways to recruit TGM who are either not yet in care or are in care and not yet on testosterone but wanting to initiate it. This will require innovative and community-engaged approaches to involving TGM who are not interfacing with the health care system in the research process, from study inception to implementation. The concept of "nothing about us without us" is crucial in ensuring not only that recruitment and enrollment of sexual and gender minority populations is fruitful, but also that the resultant science is representative of the needs of the population [22]. Potential solutions include the use of study community advisory boards, hiring key popular opinion leaders in the TGM community to facilitate trust and engagement in medical care for individuals who may be hesitant in doing so, and considering targeted social media or geosocial networking campaigns, all of which have been used to enhance engagement with other sexual and gender minority populations [23, 24]. These approaches can facilitate moving out of clinical and community-based organization settings for recruitment and into the community at large. While our team included members who were self-identified as transgender or non-binary individuals, all were rooted in medical and research institutions. Our proposed solutions are potential components of concerted efforts for deeper engagement of the transgender and non-binary community at large, which will be essential in the design of future studies.

TGM who were not yet on testosterone had difficulty screening for the study due to limited access to timely gender healthcare, including initiation of testosterone therapy. Despite 61 individuals contacting our team about this study, many of them did not present for in-person screening because of limited access to timely gender healthcare. In the Birmingham, AL metropolitan area, there are a limited number of clinics dedicated to GAHT (n = 3) and the waiting list for new patients may exceed a 6-month time period. With limited transgender healthcare resources in this area and other areas in the Southeastern US, access to affirming healthcare is an ongoing challenge for research related to the health needs of TGM and other gender diverse populations. While this was a major challenge in this study, it may present an opportunity for other types of sexual health research. Community-engaged research initiatives could serve the dual purpose of providing needed services such as STI testing and PrEP services.

Another recruitment challenge was enrollment of TGM of color and other racial/ethnic groups, particularly Black TGM. Black TGM are particularly underrepresented in all areas of health research. In the 2015 US Transgender Survey (USTS), of 27,715 participants, only 796 (2.9%) identified as Black or African American and of those, 35% identified as TGM [25]. Despite these findings, and given the limited representation of Black sexual and gender minorities in longitudinal surveys such as the USTS, the actual number of Black TGM, especially in the Southeastern US, is likely much larger. The intersecting identities (i.e., being Black as well as transgender) compound the sexual health disparities experienced by this population. Thus, engaging and recruiting Black TGM is a key area for future work.

## Variations in available GAHT routes and doses

Of those presenting to clinical sites to initiate testosterone, not all wished to be on IM or SQ testosterone regimens, as some favored transdermal preparations. The reason for selecting the IM or SQ route of testosterone administration for this study was to reduce the pharmacokinetic and pharmacodynamic variability on the genital microbiota that may be introduced by different administration routes. While there are data to support that IM and SQ testosterone

regimens result in stable serum levels [26], it is not known whether these administrations routes have variable serum levels compared to transdermal regimens. However, given the challenges in recruitment we faced in this study, inclusion of all available routes of testosterone administration should be considered in future studies to optimize recruitment when evaluating the impact of testosterone on the genital microbiota of TGM.

## Dysphoria associated with genital specimen collection

Some potential participants also experienced hesitancy toward the study's requirement of daily self-collected genital swabs for a relatively long period of time (97 days). It is important to note that our team has extensive experience conducting longitudinal genital microbiome research with multiple populations other than TGM, including cisgender women who have sex with men and cisgender women who have sex with women [27, 28]. In this field, frequent and long-term specimen collection is standard and necessary to yield rigorous results. Self-collection of genital swabs may produce feelings of gender dysphoria [29], and this needs to be balanced with the number of genital specimens needing to be collected in robust genital microbiome studies. For this study, participants were informed about the potential for dysphoria and discomfort in the collection of frequent vaginal specimens. They were instructed to reach out to the study team if they developed emotional distress related to study procedures so that mental healthcare referrals could be made. When planning for future studies in transgender populations, the frequency of study procedures that may induce dysphoria should be carefully considered to improve recruitment. Currently, data regarding the acceptability and feasibility of genital sample self-collection by TGM is limited to testing for human papilloma virus (HPV) [30], but studies in this area have shown self-collection to be acceptable and preferable to clinician collection. Qualitative studies investigating the experience of genital specimen self-collection by TGM for vaginal health evaluation are needed [30].

## Inflexible study design

One factor influencing this was the specific eligibility criteria for this genital microbiome study. As discussed above, current testosterone use for GAHT was a major reason many participants were unable to participate in screening. Given the stringent timing requirements in the study eligibility criteria, the aforementioned health care provider access limitations also made it difficult for potential participants to complete screening with subsequent timely testosterone initiation. Additionally, the requirement of waiting 7 days from screening to testosterone initiation was also a barrier for some potential participants. For many individuals, they have been waiting years to begin GAHT and in the meantime, have struggled with dysphoria and discrimination, making the waiting period difficult to justify. This was also further complicated for some individuals who were initially willing to participate, but started their menses and were not willing to delay testosterone initiation any further. Future studies with GAHT-naïve transgender populations should consider these barriers to enrollment early in the study design process. Balancing the acceptability and feasibility of study enrollment procedures with scientific objectives must be careful considered.

## Conclusion

The excitement among TGM to be represented and participate in research should be leveraged into further work which engages a variety of transgender community stakeholders from the inception of research project development, including creation of a community advisory board and engagement of diverse key opinion leaders in the community to discuss the feasibility of the study proposed and ways to mitigate recruitment challenges, should they occur. These

results also highlight the need for more clinical resources for transgender care, especially in the Southeastern US.

## Supporting information

**S1 File.**
(XLSX)

## Acknowledgments

First and foremost, we would like to thank the participants for their willingness to be a part of this research, from those who inquired about the study, to those who were screened and enrolled. We would also like to thank Dr. Scott Weisberg and his clinical staff at MCWC, Brianna Patterson at the UAB GHC, Megan Whitfield at the UAB SHC, and Brenda Ocegueda at the UAB SHRC for their recruitment support for this study. This work was presented at the 2023 STI & HIV World Congress in Chicago, IL, USA on July 25, 2023 (poster #P442).

## Author Contributions

**Conceptualization:** Olivia T. Van Gerwen, Krishmita Siwakoti, Evelyn Toh, Christopher M. Taylor, Christina A. Muzny.

**Data curation:** Z. Alex Sherman, Kristal J. Aaron, Ben Rogers.

**Formal analysis:** Olivia T. Van Gerwen.

**Funding acquisition:** Christopher M. Taylor, Nicholas J. Van Wagoner, Christina A. Muzny.

**Investigation:** Olivia T. Van Gerwen, Isaac Eastlund, Keonte J. Graves, Saralyn Richter, Angela Pontius, Evelyn Toh, Jacob H. Elnaggar.

**Project administration:** Joy Lewis, Isaac Eastlund, Keonte J. Graves, Saralyn Richter, Angela Pontius.

**Resources:** Emma Sophia Kay, Jay Wall, Krishmita Siwakoti, Nicholas J. Van Wagoner.

**Supervision:** Christina A. Muzny.

**Visualization:** Ben Rogers.

**Writing – original draft:** Olivia T. Van Gerwen.

**Writing – review & editing:** Olivia T. Van Gerwen, Emma Sophia Kay, Kristal J. Aaron, Jacob H. Elnaggar, Nicholas J. Van Wagoner, Christina A. Muzny.

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
