## [Decision Letter · Decision Letter 0]

17 May 2024

PONE-D-23-42596Challenges in recruiting transgender men in the Southeastern United States for genital microbiome researchPLOS ONE

Dear Dr. Van Gerwen,

Thank you for submitting your manuscript to PLOS ONE. After careful consideration, we feel that it has merit but does not fully meet PLOS ONE’s publication criteria as it currently stands. Therefore, we invite you to submit a revised version of the manuscript that addresses the points raised during the review process.

The strength of this paper lies in the provision of a narrative which can guide future research design. It is obvious that this was not a planned publication but "lessons learned" publications are useful to science. However, the narrative needs to be re-structured to make for easy reference for other researchers. I encourage the authors to consider a tabular presentation of the challenges and there suggested approach to overcome each one. Further, they should review the literature and present approaches used by similar study designs to overcome the challenges identified.

The work needs MAJOR revisions.

We look forward to receiving your revised manuscript.

Kind regards,

Daniel Antwi-Amoabeng, MD, MSc

Academic Editor

PLOS ONE

2. Thank you for stating the following financial disclosure: "OTVG has received funding to her institution for research from Gilead Sciences, Inc., Moderna Pharmaceuticals, Visby, and Abbott Molecular and has served on a scientific advisory board for Scynexis, for which she received honoraria. CAM reports receiving grants to her institution from NIAID, Lupin, Abbott Molecular, Visby, and Gilead Sciences, Inc. She also reports honorarium and/or consulting fees from Scynexis, Cepheid, BioNTech, Visby Medical, Elsevier, UpToDate, Abbott Molecular, and Roche. All other authors have no disclosures.

This study is funded by the National Institute of Allergy and Infectious Diseases (R21AI167754-01 granted to CAM, NVW, and CMT). OTVG is currently funded by a 2022-2024 Frommeyer Career Development Award from the UAB Department of Medicine." 

3. In the online submission form, you indicated that the data which are not available within the manuscript and its supporting information files is available per written request to the corresponding authors. It is currently being stored and managed in a local redcap database and must be deidentified to be shared.

Introduction: well-written.

Methods:

What was the rationale for the required 3 daily genital swab collection and the daily sexual behavior journal?

There were going to be at least 6 trips to deposit study materials. This is an onerous task even for a research team, much more so for a participant.

Does the data need to be this granular? I suspect the intensity of the data collection may have contributed to folks declining to participate.

Table 1, line 1 of the “exclusion criteria” column, please edit to read “...use in the preceding 14 days.”

The IRB approval expired on 26-Aug-2022. Please state if an extension was sought to allow recruitment past that date and provide documentation to support this.

Results:

Line 143: the second sentence should be in the past tense.

Line 144: Please state why enrollment stopped.

Discussion and conclusion:

How did this recruitment plan differ from other studies involving this patient population? Are there any studies with less stringent criteria but had robust data?

Reviewers' comments:

Reviewer's Responses to Questions

**Comments to the Author**

1. Is the manuscript technically sound, and do the data support the conclusions?

Reviewer #1: Partly

Reviewer #2: Yes

2. Has the statistical analysis been performed appropriately and rigorously? 

Reviewer #1: No

Reviewer #2: N/A

3. Have the authors made all data underlying the findings in their manuscript fully available?

Reviewer #1: Yes

Reviewer #2: No

4. Is the manuscript presented in an intelligible fashion and written in standard English?

Reviewer #1: No

Reviewer #2: Yes

5. Review Comments to the Author

Reviewer #1: Throughout the manuscript the tenses keep changing. Some parts seem as though the study is still ongoing. Is the study ongoing?

In the background the authors should clearly write the aim of the study

The methodology should be clearly written

What was the study design? I notice narratives in the method section

What tools were used for analysis?

In table 2 the terms used under gender identity should have been defined earlier in the text

What is the strength of this study?

What are the limitations of this study

Most part of the conclusion would seem to be a discussion. The authors need to clearly write the conclusion their findings

Reviewer #2: The paper by Van Gerwen et al. on “Challenges in recruiting transgender men in the Southeastern United States for genital microbiome research” will make a great addition to the literature. I appreciate that you highlight the cultural competency of the study team and community engagement in the planning of this study. This article could serve as a guide to other researchers in the field of transgender health research. I have provided my comments below.

1. Participant recruitment through testosterone providers – this would make it hard to find people not already on testosterone. Could you have contacted people on the wait list?

a. How well respected are these providers in the community? Do they feel like safe and welcoming spaces to TGM?

2. Could you talk about why you required 3 daily vaginal swabs? Why would 1 per day or 1 per week not suffice?

a. Of the enrolled individuals, how many returned all the swabs? What proportion were returned? Did the number of returned swabs decrease over the study period?

b. What collection device was provided to participants? What was the rationale for choosing that particular device? Was comfort to participants considered?

c. Are you collecting feedback from participants on what has been helpful or determinantal to vaginal collection?

3. Could you provide more detail on the research question and hypotheses under study? This will help the reader understand why so many samples are needed and the rationale behind the eligibility criteria.

a. Related, could you briefly provide details about the rationale behind the inclusion/exclusion criteria? For instance, please describe the Amsel criteria and Nugent score.

4. What is the target sample size you were aiming for?

5. When more than 1 reason was cited by people for non-participation, how did you determine the most important reason? It would be interesting to note multiple barriers to participation.

6. On lines 192-194 you say, “Concerted efforts for deeper engagement of the transgender and non-binary community at large will be essential in the design of future studies.” Later, you mention forming a community advisory board. Could you talk about this a bit more in the same paragraph as lines 192-194? Who would be included and what do you think you would gain? Consulting with community early and often may have increased participation and could be presented a lesson learned to encourage other researchers to put together a CAB before starting recruitment.

7. You speak about the lack of Black TGM individuals in your study. Are there any Black LGBTQ+ individuals on your study team who could help you reach out to this population?

8. What is happening to the study? Are the 10 individuals you enrolled adequate to complete the study aims? Or will you have another recruitment round after working more closely with the community?

6. PLOS authors have the option to publish the peer review history of their article (what does this mean?). If published, this will include your full peer review and any attached files.

Reviewer #1: No

Reviewer #2: No

---

## [Author Response · Author response to Decision Letter 0]

19 Jun 2024

June 17, 2024

Subject: Responses to reviewers for manuscript PONE-D-23-42596

Dear Editors,

We appreciate the thoughtful comments from the reviewers for our article entitled, “Recruiting transgender men in the Southeastern United States for genital microbiome research: lessons learned.” We have addressed their critiques as detailed below and resubmitted the updated manuscript (both clean and tracked copies). Line numbers below refer to the clean version. 

1. The strength of this paper lies in the provision of a narrative which can guide future research design. It is obvious that this was not a planned publication but "lessons learned" publications are useful to science. 

Author response: Thank you to the editor for this comment. We agree that the “lessons learned” approach is the most useful way of presenting our results. We have changed our title to “Recruiting transgender men in the Southeastern United States for genital microbiome research: lessons learned” in our response.

2. However, the narrative needs to be re-structured to make for easy reference for other researchers. I encourage the authors to consider a tabular presentation of the challenges and there suggested approach to overcome each one. 

Author response: We have revised the discussion section to reflect more of a lessons learned format. We have also created a new table displaying several of the challenges we have encountered and how best to overcome them in future projects (new Table 3 in the revised manuscript). Briefly, our table displays four major categories of challenges (which are also described in detail in the text of the discussion section): limited engagement with TGM not in care, variations in available GAHT routes and doses, dysphoria associated with genital specimen collection, and inflexible study design. We propose some potential solutions that can be used by other investigators looking to engage this population in future studies, including several community-engaged research strategies which have been used successfully with other sexual and gender minority populations. (Pages 9-14)

3. Further, they should review the literature and present approaches used by similar study designs to overcome the challenges identified.

Author response: To our knowledge, this is the first prospective vaginal microbiome/BV pathogenesis study in TGM. We have added text to the revised discussion section regarding lessons learned from research in adjacent fields engaging this population (e.g., HIV research, surgical research), however, BV pathogenesis research itself is quite unique in its nuances and complexity. Thus, the challenges presented in this work are difficult to directly compare to these other fields. Nevertheless, we feel there are lessons to be learned from general community engagement principles and their utility in research involving TGM, therefore we have included some of these lessons learned and future directions in the revised discussion section and the new Table 3 that we have also created. 

4. Thank you for stating the following financial disclosure: "OTVG has received funding to her institution for research from Gilead Sciences, Inc., Moderna Pharmaceuticals, Visby, and Abbott Molecular and has served on a scientific advisory board for Scynexis, for which she received honoraria. CAM reports receiving grants to her institution from NIAID, Lupin, Abbott Molecular, Visby, and Gilead Sciences, Inc. She also reports honorarium and/or consulting fees from Scynexis, Cepheid, BioNTech, Visby Medical, Elsevier, UpToDate, Abbott Molecular, and Roche. All other authors have no disclosures. This study is funded by the National Institute of Allergy and Infectious Diseases (R21AI167754-01 granted to CAM, NVW, and CMT). OTVG is currently funded by a 2022-2024 Frommeyer Career Development Award from the UAB Department of Medicine." Please state what role the funders took in the study. If the funders had no role, please state: ""The funders had no role in study design, data collection and analysis, decision to publish, or preparation of the manuscript."" If this statement is not correct you must amend it as needed. 

Author Response: This has been updated to reflect that the funder had no role in these activities. Consider this the Role of Funder statement in our cover letter as well. We have also updated the financial disclosures of senior author CAM. 

5. In the online submission form, you indicated that the data which are not available within the manuscript and its supporting information files is available per written request to the corresponding authors. It is currently being stored and managed in a local redcap database and must be deidentified to be shared. All PLOS journals now require all data underlying the findings described in their manuscript to be freely available to other researchers, either 1. In a public repository, 2. Within the manuscript itself, or 3. Uploaded as supplementary information. This policy applies to all data except where public deposition would breach compliance with the protocol approved by your research ethics board. If your data cannot be made publicly available for ethical or legal reasons (e.g., public availability would compromise patient privacy), please explain your reasons on resubmission and your exemption request will be escalated for approval. 

Author Response: We have now created a clean, deidentified table of our recruitment data and submitted it as a supplemental document in our revised submission. Of note, the originally submitted manuscript numbers have been updated to reflect the final results of those who inquired about the study, were screened, and were enrolled as the study was still ongoing at the time of initial submission to PLoS One (the study is now closed and in data analysis only).

6. What was the rationale for the required 3 daily genital swab collection and the daily sexual behavior journal? 

Author Response: As detailed in our published protocol paper for this study (reference #10, Muzny, et al, BMJ Open, 2023), the three daily genital swabs and daily diaries are necessary to comprehensively study the longitudinal impact of GAHT on the composition of the genital microbiome over time. We have added additional information to the revised manuscript to explicitly state that each swab is used for a specific purpose that meets the aims of the parent study (Page 6, Lines 149-151), including one swab for Nugent score determination, one swab for 16S rRNA gene sequencing, and one swab for shotgun metagenomic sequencing. Regarding the daily diaries, events such as sex, douching, and menses can have a major impact on the composition of the genital microbiome and this can shift from day to day, requiring frequent collection of these data. We have added a sentence to the revised manuscript to emphasize this point (Page 6, Lines 152-155). Overall, the methods used in this study are standard in the field of vaginal and genital microbiome research (particularly BV pathogenesis research), including work which has been done by this team with other populations (PMID 30872415, PMID: 38316599, PMID: 24451163).

7. There were going to be at least 6 trips to deposit study materials. This is an onerous task even for a research team, much more so for a participant.

Author Response: We recognize the frequency of returning study materials is asking a lot of participants. However, given the volume of samples and the workflow required for processing of samples, frequent drop-offs to the clinical site were needed. We did allow for every two-week drop-offs if participants required such accommodations. Participants were also compensated at each drop-off visit in addition to the compensation they received at their initial screening and enrollment visits, as detailed on Page 7, 155-158. As noted in response #6 above, frequent specimen collection is a component of longitudinal genital microbiome studies, including those on BV pathogenesis and has been used in prior studies. 

8. Does the data need to be this granular? I suspect the intensity of the data collection may have contributed to folks declining to participate.

Author Response: As noted above in response #6 and #7, these methods are standard for longitudinal vaginal/genital microbiome research. We address the frequency, as well as other challenges associated with specimen collection in the discussion section of the revised manuscript (Page 13, Lines 271-289) 

9. Table 1, line 1 of the “exclusion criteria” column, please edit to read “...use in the preceding 14 days.”

Author Response: This has been changed in the revised Table 1.

10. The IRB approval expired on 26-Aug-2022. Please state if an extension was sought to allow recruitment past that date and provide documentation to support this.

Author Response: Annual continuing IRB approval has been granted for every year of this study since our team received the R21 funding award from NIAID. A copy of the most recent annual continuing review letter is included with this revised submission. Please let us know if you require additional information on this topic. 

11. Line 143: the second sentence should be in the past tense.

Author Response: We have edited the manuscript to be consistently in past tense throughout.

12. Line 144: Please state why enrollment stopped.

Author Response: We have added a sentence addressing this- enrollment stopped because the 2 year R21 grant funding the study ended. (Page 8, Lines 168-169)

13. How did this recruitment plan differ from other studies involving this patient population?

Author Response: This is the first prospective study engaging TGM in genital microbiome research, so there are no standards for how to recruit this particular population for this particular type of research. Community-engaged methods that have been successfully used by our research team to engage other sexual and gender minority groups in both general sexual health research as well as genital microbiome research were utilized. The recruitment methods used are described in detail in the manuscript on Pages 4 and 5. 

14. Are there any studies with less stringent criteria but had robust data?

Author Response: As noted above in response #13, this is the first prospective study enrolling TGM to investigate microbiome impacts of any kind involving the genital mucosa. Thus, we are unable to directly compare our experience in this study with other studies. 

15. Throughout the manuscript the tenses keep changing. Some parts seem as though the study is still ongoing. Is the study ongoing?

Author Response: At the time the manuscript was written and submitted to PLoS One (end of 2023), we were just finishing long-term follow-up on several participants. The study closed to enrollment on 10/5/23 and participants completed their follow-up procedures by the end of 2023. Currently the study is in data analysis only. This is now clarified in the revised manuscript. Additionally, the manuscript has been converted to past tense throughout to avoid any confusion on this topic. 

16. In the background the authors should clearly write the aim of the study.

Author Response: We have augmented the final sentence of the revised introduction to clearly state our study aim of presenting our team’s recruitment strategies for this study, the socio-demographics of both screened and enrolled participants, and describe the lessons learned through our recruitment challenges to help inform the design of future studies engaging underserved populations such as TGM in genital microbiome research. (Page 4, Lines 98-100)

17. The methodology should be clearly written.

Author Response: The methods for this study are published in a protocol paper that is extensively cited throughout this manuscript (reference #10). We have described our recruitment and engagement methods in this paper extensively on Pages 4-6.

18. What was the study design? I notice narratives in the method section.

Author Response: The methods for this study are published in a protocol paper that is extensively cited throughout this manuscript. (reference #10) We have described our recruitment and engagement methods in this paper extensively on Pages 4-6.

19. What tools were used for analysis?

Author Response: The data analysis approach for the parent study is described in depth in the cited study protocol paper (reference #10). Regarding how we analyzed the recruitment data presented in this paper, we have added a short section to the revised methods describing the recording of descriptive recruitment data in an Microsoft Excel spreadsheet throughout the study period and the calculation of the descriptive statistics presented in the results section using Microsoft Excel. (Page 6, Lines137-145) 

20. In table 2 the terms used under gender identity should have been defined earlier in the text

Author Response: We have now expanded mention of the gender identities listed in the inclusion criteria section of the revised Table 1. These identities all fall under the “transgender and gender diverse” umbrella.

21. What is the strength of this study?

Author Response: The strengths of this study are described in the Discussion section in the first paragraph. Specifically, to our knowledge, this is the first study to prospectively enroll TGM in the Southeastern US to investigate the genital microbiota pre- and post-testosterone initiation, including the development of BV. We utilized some community engagement methods that were successful in yielding 18 screening visits (one patient was screened twice) and 10 enrolled participants and these are also described in detail in both the methods and the discussion sections of the manuscript. Another strength of this study is our description of the lessons learned from our recruitment challenges, which have allowed our team to identify areas of opportunity in improving TGM recruitment and engagement in future research studies. These strengths are now included in the new Table 3 and throughout the Discussion section. (Pages 9-13)

22. What are the limitations of this study?

Author Response: Limitations of this study are described in the Discussion section. They are also shown in the new Table 3 as the challenges we encountered with recruiting and enrolling for this study.

23. Most part of the conclusion would seem to be a discussion. The authors need to clearly write the conclusion their findings

Author Response: The Discussion has now been edited to be more structured and focused on the lessons learned and potential opportunities for future research with this population in this scientific realm. In the revised manuscript, a conclusion section after the discussion now summarizes these themes and provides concluding thoughts about how teams can move forward when proposing similar studies with this population. 

24. Participant recruitment through testosterone providers – this would make it hard to find people not already on testosterone. Could you have contacted people on the wait list?

Author Response: As described in the parent study protocol paper (reference #10), participants had to undergo study procedures for the 7 days prior to testosterone initiation, then receive their first testosterone injection, and then be followed for 90 days. Crucially, our research team was not capable of nor responsible for prescribing testosterone for research participants. This had to be done in a clinical, non-research setting. Therefore, contacting people off of the clinic wait lists would not have provided us with any additional participants because they would still have to wait to engage with their testosterone provider.

25. How well respected are these providers in the community? Do they feel like safe and welcoming spaces to TGM?

Author Response: All providers and clinical care sites participating in this study are well-respected by the local transgender community. They have also all gone through cultural competency trainings, which have yielded warm and welcoming spaces where transgender patients feel safe and celebrated. We have added a line to the manuscript to clarify this. (Lines 108-11). Outside of the clinical partners involved in this study, the UAB Sexual Health Research

---

## [Decision Letter · Decision Letter 1]

29 Jul 2024

Recruiting transgender men in the Southeastern United States for genital microbiome research: lessons learned

PONE-D-23-42596R1

Dear Dr. Van Gerwen,

We’re pleased to inform you that your manuscript has been judged scientifically suitable for publication and will be formally accepted for publication once it meets all outstanding technical requirements.

Kind regards,

Daniel Antwi-Amoabeng, MD, MSc

Academic Editor

PLOS ONE

Additional Editor Comments (optional):

The authors have addressed all reviewer questions. The work is accepted for publication.

Reviewers' comments:

Reviewer's Responses to Questions

**Comments to the Author**

1. If the authors have adequately addressed your comments raised in a previous round of review and you feel that this manuscript is now acceptable for publication, you may indicate that here to bypass the “Comments to the Author” section, enter your conflict of interest statement in the “Confidential to Editor” section, and submit your "Accept" recommendation.

Reviewer #1: All comments have been addressed

Reviewer #2: All comments have been addressed

2. Is the manuscript technically sound, and do the data support the conclusions?

Reviewer #1: Yes

Reviewer #2: Yes

3. Has the statistical analysis been performed appropriately and rigorously? 

Reviewer #1: Yes

Reviewer #2: N/A

4. Have the authors made all data underlying the findings in their manuscript fully available?

Reviewer #1: Yes

Reviewer #2: Yes

5. Is the manuscript presented in an intelligible fashion and written in standard English?

Reviewer #1: Yes

Reviewer #2: Yes

6. Review Comments to the Author

Reviewer #1: All comments have been responded to and the manuscript, these include improvement in the English language, it may proceed to the next step

Reviewer #2: (No Response)

7. PLOS authors have the option to publish the peer review history of their article (what does this mean?). If published, this will include your full peer review and any attached files.

Reviewer #1: **Yes: **ESTER ACEN

Reviewer #2: No

---

## [Editor Report · Acceptance letter]

2 Aug 2024

PONE-D-23-42596R1 

PLOS ONE

Dear Dr. Van Gerwen, 

I'm pleased to inform you that your manuscript has been deemed suitable for publication in PLOS ONE. Congratulations! Your manuscript is now being handed over to our production team.

Kind regards, 

on behalf of

Dr. Daniel Antwi-Amoabeng 

Academic Editor

PLOS ONE